# Protein fractional synthesis rates within tissues of high- and low-active mice

**Kristina M. Cross**[1]*, **Jorge Z. Granados**[2], **Gabriella A. M. Ten Have**[1], **John J. Thaden**[1], **Marielle P. K. J. Engelen**[1], **J. Timothy Lightfoot**[2‡], **Nicolaas E. P. Deutz**[1‡]

**1** Center for Translational Research in Aging & Longevity, Dept. Health and Kinesiology, Texas A&M University, College Station, TX, United States of America, **2** Biology of Physical Activity Laboratory, Dept. Health and Kinesiology, Texas A&M University, College Station, TX, United States of America

‡ These authors are joint senior authors on this work.
* km.cross@ctral.org

**Data Availability Statement:** All relevant data are within the manuscript.

**Funding:** The authors received no specific funding for this work.

## Abstract

With the rise in physical inactivity and its related diseases, it is necessary to understand the mechanisms involved in physical activity regulation. Biological factors regulating physical activity are studied to establish a possible target for improving the physical activity level. However, little is known about the role metabolism plays in physical activity regulation. Therefore, we studied protein fractional synthesis rate (FSR) of multiple organ tissues of 12-week-old male mice that were previously established as inherently low-active (n = 15, C3H/HeJ strain) and high-active (n = 15, C57L/J strain). Total body water of each mouse was enriched to 5% deuterium oxide ($D_2O$) *via* intraperitoneal injection and maintained with $D_2O$ enriched drinking water for about 24 h. Blood samples from the jugular vein and tissues (kidney, heart, lung, muscle, fat, jejunum, ileum, liver, brain, skin, and bone) were collected for enrichment analysis of alanine by LC-MS/MS. Protein FSR was calculated as -ln(1-enrichment). Data are mean±SE as fraction/day (unpaired t-test). Kidney protein FSR in the low-active mice was 7.82% higher than in high-active mice (low-active: 0.1863±0.0018, high-active: 0.1754±0.0028, p = 0.0030). No differences were found in any of the other measured organ tissues. However, all tissues resulted in a generally higher protein FSR in the low-activity mice compared to the high-activity mice (e.g. lung LA: 0.0711±0.0015, HA: 0.0643±0.0020, heart LA: 0.0649± 0.0013 HA: 0.0712±0.0073). Our observations suggest that high-active mice in most organ tissues are no more inherently equipped for metabolic adaptation than low-active mice, but there may be a connection between protein metabolism of kidney tissue and physical activity level. In addition, low-active mice have higher organ-specific baseline protein FSR possibly contributing to the inability to achieve higher physical activity levels.

## Introduction

It has been shown repeatedly that physical activity level has both genetic and biological regulating factors [1]. Previously, it has been demonstrated that inherently high-active mice (C57L/J), when compared to an inherently low-active mouse strain, (C3H/HeJ) have overexpression of

**Competing interests:** The authors have declared no competing interests exist.

proteins associated with metabolism in the nucleus accumbens of the brain [2] and in skeletal muscle [3]. The observed overexpression and inferred upregulated metabolism of these proteins support the hypothesis that the capacity to increase protein synthesis in specific organs may be related to the physical activity level of an organism. Therefore, in search of what drives an organism to be more active, we hypothesized that there are differences in tissue protein synthesis between animals displaying differing physical activity levels.

Protein synthesis is commonly measured by calculating the protein fractional synthesis rate (FSR), which is defined as the rate that labeled amino acid precursor is incorporated into protein. Protein FSR gives insight into metabolic regulation because the ability to create proteins faster represents the capability to adapt quicker by upregulation or downregulation of processes controlled by protein concentrations. For instance, the intestines have a high protein FSR [4] which allows for the intestinal system to quickly react to stimuli or environmental changes.

A generally accepted method of measuring protein FSR is the administration of heavy water ($D_2O$) to create labeled amino acids [5]. This method has been used to measure tissue-specific protein FSR of skeletal muscle [6], heart, and liver [7]. While other organ tissues have been analyzed for protein FSR utilizing other stable isotope methods (i.e. primed continuous intravenous infusions with L-[ring-$^{13}C_6$]-Phenylalanine) [8], in the present study, we utilized $D_2O$ to measure protein FSR in several organ tissues simultaneously as a multi-organ approach to examine the association between protein metabolism and physical activity level. Therefore, we measured protein FSR of inherently high- and low-active inbred mouse strains (C57L/J and C3H/HeJ, respectively) [3, 9–11].

## Methods

**Animals.** All procedures were approved by the Texas A&M University Institutional Animal Care and Use Committee (IACUC; AUP # 2015–0159). We analyzed samples from 15 male C3H/HeJ mice (inherently low-active inbred strain) and 15 male C57L/J mice (inherently high-active inbred strain), purchased from The Jackson Laboratory (Bar Harbor, ME, USA). Average physical activity levels of these two mouse strains have previously been described [12]. In previous studies, we observed that low-active mice and high-active mice have daily wheel running on average of 0.6 ± 1.1 km/day and 9.5 ± 2.0 km/day (mean±SD), respectively [12]. This physical activity level is shown to be highly reproducible [13]. We used C57Bl6/J mice (n = 5), to estimate tracer background enrichments. All mice were obtained at 10-weeks of age and group-housed in standard mouse-cages in a light and temperature-controlled housing facility (12-hour light-dark cycle, room temperature 22–24˚C) with *ad libitum* water and a standard chow diet (Harlan Labs, Houston TX; 25.2% protein, 4.0% fat, 39.5% carbohydrate, 3.3% crude fiber, 10% neutral fiber, and 9.9% ash). Food consumption was measured daily by measuring unconsumed food weights. The food intake responses between high and low active mice has been shown previously [11]. During the two-week acclimation period, mice were not exposed to wheel-running activity because we previously showed that multiple day exposure to running wheels can result in gene expression changes and potential alteration of protein FSR and protein expression [10]. By studying the mice strains without the physical activity stimulus of running wheel activity, we are able to analyze potential differences in protein metabolism that may be driving activity level while preventing the potential confounding of the protein FSR/protein expression responses by physical activity exposure.

## Study protocol

**$D_2O$ administration.** After the acclimation period, we performed the protein FSR assessment procedures *via* a terminal surgery. The day before tissue collection, the mice were given

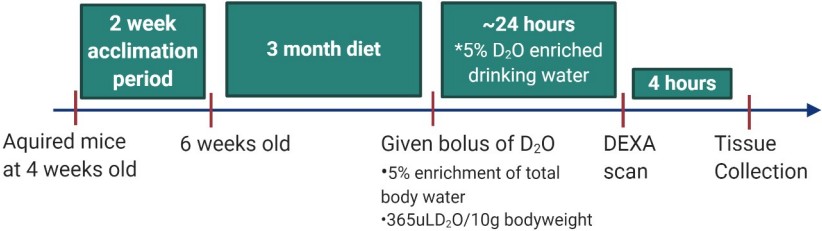

**Fig 1. Timeline of study.**

an intraperitoneal injection of a pre-calculated amount of $D_2O$ (365uL/10g of body weight) to raise body water enrichment to a level of 5%. The $D_2O$ will enrich the amino acid alanine for calculation of the protein FSR at a sufficient precision in all the organs of interest [14, 15] and is sufficient to achieve a measurable amount of labeled alanine in the tissues with the lowest and the highest predicted protein FSR. After the $D_2O$ injection, the mice were given the same percentage of $D_2O$ in their drinking water to maintain steady-state $D_2O$ enrichment for ~24 hours until the start of the surgical procedure. The timeline of the study is shown in Fig 1.

**Body composition.**   Four hours before initiation of the terminal surgical procedure, food was removed from mouse cages to study the animals in the post-absorptive condition. Body weight (bw) was assessed immediately after food withdrawal using a digital beam scale. Lean body mass, and fat masswere measured using echo MRI (EchoMRI LLC, Houston, TX 77079). Total body water was estimated by calculating 74% of body weight [16].

**Anaesthesia.**   With the goal of this study to answer what regulates physical activity level, we analyzed multiple aspects of metabolism. This study specifically reports our findings of protein FSR while findings regarding plasma amino acid kinetics obtained by the administration of stable isotope tracers are reported in Granados et. al. [17]. Due to the additional measurements, surgery was necessary and blood samples were taken along the timeline of the greater study. We anaesthetized the mice with a mixture of ketamine (1.25 mg/10g BW) and medetomidine (2 μg/10g BW) *via* intraperitoneal injection (0.1 ml/10g bw) [18] and maintained anaesthesia using a continuous subcutaneous pump infusion of ketamine (0.35 mg/ 10 g BW/h) and medetomidine (0.35 μg/10g BW/h) at a rate of 0.1 ml/10 g BW/h [18]. Under anaesthesia, a peripheral catheter was placed in the right jugular vein for blood sampling. We maintained the fluid balance and blood pressure by an initial 1.5 ml IP saline injection (0.9% sterile, NaCl), and by continuous subcutaneous pump infusion (Harvard PHD2000) of normal saline at a rate of 2.5 ml/hour [18]. Breathing was continuously monitored and body temperature was maintained at 37°C using a heating pad and lamp.

**Tissue collection and processing.**   Blood samples (0.05–0.1 ml per sample) were collected on two sampling time schedules (schedule 1: t = 3 and 30 minutes; schedule 2: t = 5 and 40 minutes) as previously described [17]. Blood samples were replaced with an equal volume of sterile normal saline. We collected blood in lithium-heparinized tubes (microtube; Sarstedt, Newton, NC) and immediately placed them on ice (4°C) then centrifuged (4 °C, 3120 x *g* for 5 min) the samples to obtain plasma within the hour. The plasma was deproteinized with 0.1 vol of 33% (w/w) trichloroacetic acid and stored at −80 °C for later analysis.

Eleven tissues were collected from each mouse in the following order: skeletal muscle, fat (peri-testicular), jejunum, ileum, liver, kidney, lung, heart, brain, skin (ear), and bone (skull). Each tissue was quickly freeze clamped, snap-frozen in liquid nitrogen, and stored in -80°C. Before analysis, the tissue homogenizing and processing was done as described [4]. In brief, the homogenized tissue powder (30–80 mg) was centrifuged to separate the supernatant, which was used for intracellular amino acid concentration measurements, and the protein

pellet. The pellet was washed three times with 0.3M TCA and hydrolyzed with 1 mL of 6 N HCl at 110°C for 24 h. 5 mL of water was added.

## Alanine enrichment analysis

**Plasma.** Plasma samples were chemically derivatized before liquid chromatography-tandem mass spectrometry (LC-MS/MS) by a modification for small plasma volumes of a method previously described [17, 19]. In brief, deproteinized plasma supernatant was reacted at room temperature for 15 min in a mixture of 3 vol of 1 M sodium phosphate, pH 10, and 6 vol of a 5% (w/v) solution in acetonitrile of 9-fluorenylmethoxycarbonyl (Fmoc) chloride (Sigma Aldrich, St. Louis MO, USA). Reactions were stopped by adding 3 vol of 1.3 M acetic acid in 8% acetonitrile. A 1.3 µL sample was metered into a 24 µL/min eluent flow (Eksigent, a part of SCIEX, Foster City CA, USA) thus onto a 0.5 x 100 mm LC column packed with 2.7 µm Halo$_®$ C18 beads (Advanced Materials Technology, Wilmington DE, USA). Fmoc derivatized alanine and heavy (mass+1) alanine eluted after 45 s as acetonitrile increased from 26.5% to 38.5% (v/v), displacing 11.6 µM ammonium acetate and 3.9% 2-propanol. Detection was by Turbo-V electrospray MS/MS on a 5500 QTRAP system (SCIEX), monitoring m/z 310 > 88 and m/z 311 > 89 at collision energies (CE) -68 V and -29 V for the natural and mass+1 alanine, respectively. Collision energy (CE) was optimized in the alanine mass+1 MS/MS channel, but deoptimized for alanine M0 itself. This allowed more samples to be injected for better precision while measuring low-level mass+1 alanine species while avoiding saturation of the ion-counting detector by the much more abundant alanine species. As the same "detuning" was applied for external-standard calibration samples, the modeled calibration line removes the detuning factor.

**Tissue intracellular fluid.** The intracellular fluid portion of the tissue homogenate was Fmoc-reacted and chromatographed in the standard manner for plasma [19], except the concentration of acetonitrile in the reaction stop solution was decreased from 8% to 4% and the volume injected for LC was reduced to 75 nL. The 310 > 88 and m/z 311 > 89 channels were monitored as above, except at CE -60 V and -29 V, respectively.

**Tissue protein.** The protein-bound portion of the tissue homogenate, the diluted acid hydrolyzate, was Fmoc-reacted and chromatographed as for plasma [19], except the reaction borate buffer was made more basic by adding 4.286 mL of 6M NaOH per 50 mL. The reaction stop solution was composed of 12 mL of acetonitrile and 5.72 mL of glacial acetic acid adjusted to 100 mL with water. 125 nL were injected and collision energies were set at -68 V and -29 V, respectively.

**Calibration of alanine enrichment.** An isotope response ratio was calculated for every injected sample by dividing the peak area measured in the alanine (mass+1) channel by that of alanine. From the response ratio was calculated the alanine enrichment by using a calibrated reference standard consisting of a pool of biosamples containing mass+1 alanine. Reference standard was reacted in several vials spaced among the biosamples and injected onto the LC column to control for experimental drift. In a preliminary experiment, the reference standard was calibrated against a set of enrichment standards, solutions in 0.1 M hydrochloric acid of a physiological concentration of alanine and increasing, small concentrations of alanine(mass +1) to give known enrichments 0.00086, 0.00432, 0.00865, 0.0432, 0.0865, 0.0173, and 0.432.

## Calculations

We have learned from other studies that the alanine enrichments from water labelling are an inconsistent measurement as the enrichment ratio between alanine and D$_2$O can range from 3.7 [20, 21] to 4.0 [22]. For this reason, we chose to analyze the true precursor of the

incorporation into protein alanine rather than an indirect measurement. By calculating the tracer-to-tracee ratio (TTR), we are able to report a more accurate enrichment in the precursor pool. We calculated the labeled to unlabeled ratio (tracer-tracee ratio = TTR = enrichment) of alanine of each organ tissue and corrected the value with the natural abundance of the tracer, measured in tissues of C57Bl6/J mice that were not given $D_2O$. Using the corrected TTR, we calculated protein FSR using Eq (1) where $f$ is calculated as protein-bound alanine (pbALA) TTR divided by the precursor pool alanine TTR [15, 23, 24]. $f$ will be between 0 and 1. The precursor pool can be measured in either the intracellular fraction of the tissue or in plasma, assuming that the plasma alanine (plaALA) TTR is the overall mean of the precursor TTR as all pools will be labeled from the $D_2O$ at the same enrichment [25]. The amount of time ($t$) is the duration between the bolus administration of $D_2O$ and the collection of tissue.

$$FSR = \frac{-\ln(1-f)}{t} \qquad (1)$$

## Statistical analysis

Results are expressed as mean [95% confidence interval]. GraphPad Prism (Version 8.3.1) was used for data analysis. If data failed normality using Graphpad Prism's normality tests, they were log-transformed before statistical tests were performed and rechecked for normal distribution. Unpaired t-tests were used to determine significant differences of the characteristics between the high and low active mice, the protein FSR of each organ, and the difference in tissue protein FSR between the high-active and low-active mice groups with the level of significance alpha set a priori at $< 0.05$. The p-values from the t-test were analyzed using false discovery rate (FDR) approach controlled with Q = 0.05 and set to use the two-stage linear step-up procedure of Benjamini, Krieger, and Yekutieli [26]. FDR adjusts the p-values to recognize if the significance is a true discovery ($q < 0.05$). The reported coefficient of variation is an output of Prism calculated as the standard deviation divided by the mean, expressed in percentage. All mean differences are calculated as the percentage of 1 minus the ratio of the two means.

# Results

## Animal characteristics

We analyzed a total of 30 male mice (15 low-active, and 15 high-active) at 12 weeks of age (Table 1). The high-active mice were characterized by 7.0% higher total body weight (p<0.0001) with a 7.6% higher lean mass (p<0.0001); however, we found no differences in total fat mass (p = 0.3868) or average daily food consumption (p = 0.5552). In addition, the high-active mice had 7.0% higher total-body water.

**Table 1. Characteristics of mice.**

| Characteristic | High-active mice (n = 15) | Low-active Mice (n = 15) | p-value |
|---|---|---|---|
| Bodyweight (g) | 26.98 [26.52, 27.44] | 25.22 [24.56, 25.88] | <**0.0001**\* |
| Lean Mass (g) | 21.91 [21.50, 22.31] | 20.36 [19.95, 20.77] | <**0.0001**\* |
| Fat Mass (g) | 2.37 [2.04, 2.71] | 2.33 [2.03, 2.63] | 0.3868 |
| Avg. Daily Food Consumption (g) | 3.17 [2.75, 3.59] | 3.39 [2.72, 4.06] | 0.5552 |
| Total-body water estimate (g) | 20.15 [19.81, 20.50] | 18.84 [18.35, 19.33] | <**0.0001**\* |

Note: unpaired t-test, Data are expressed as mean [95% CI] in grams

\*significance (p<0.05)

## Plasma vs. Intracellular alanine TTR for calculating fractional synthesis rates

Preliminary to calculating protein FSR, the precursor pool was determined since either plasma or intracellular alanine TTR values can be used to calculate protein FSR. We measured alanine TTR within plasma at various blood sampling time points: 3 minutes and 30 minutes, or 5 minutes and 40 minutes. Since the values of alanine TTR at the early compared to the later time points are not statistically different (3 min: 0.1962 [0.1876, 0.2049] versus 30 min: 0.1964 [0.1876, 0.2051] p = 0.8181, and 5 min: 0.2044 [0.2014, 0.2075] versus 40 min: 0.2054 [0.2023, 0.2085] p = 0.6741), the plasma is a stable measurement to use as the precursor pool for protein FSR calculations. Also the calculated coefficient of variation showed low variability change between the initial time points (3 min: 8.46%, 5 min: 2.53%) and later time points (30 min: 7.82%, 40 min: 2.40%). Alanine TTR within the intracellular fraction of tissue (icALA TTR) resulted in more variability when compared to the plasma alanine (plaALA) TTR as shown by the coefficient of variation (Table 2). In addition, when using the plaALA TTR or icALA TTR as the precursor pool to calculate protein FSR, we observed more variability in the protein FSR calculated with icALA TTR in each organ (Table 3). The coefficients of variation were two-fold higher in some tissues when using icALA TTR to calculate the protein FSR. Also, the protein FSRs calculated with icALA TTR were significantly higher than when calculated with plaALA TTR in most tissues (Table 3). Since the icALA TTR were more variable than the plaALA TTR, we, therefore, decided to use the plaALA TTR as the precursor pool to calculate protein FSR.

## Tissue protein fractional synthesis rates

Utilizing the plasma alanine TTR (Fig 2A), the protein-bound alanine TTR (Fig 2B), and sampling time to calculate protein FSR of the mice given 5% $D_2O$ enrichment, we found the jejunum had the highest tissue protein FSR followed by ileum and liver while skeletal muscle, bone and skin had the lowest protein FSR (Fig 3).

**Table 2. Comparison of the precursor pools used to calculate FSR for all mice combined.**

|  | Alanine TTR (n = 30) | Coefficient of Variation (%) |
|---|---|---|
| **PlaALA TTR** |  |  |
| Plasma | 0.2013 [0.1968, 0.2058] | 5.96% |
| **Tissue icALA TTR** |  |  |
| Kidney | 0.1599 [0.1461, 0.1738] | 22.72% |
| Heart | 0.1720 [0.1664, 0.1776] | 7.89% |
| Fat | 0.1914 [0.1871, 0.1957] | 5.59% |
| Lung | 0.1722 [0.1624, 0.1819] | 14.85% |
| Jejunum | 0.1698 [0.1587, 0.1809] | 17.19% |
| Ileum | 0.1878 [0.1660, 0.2095] | 30.51% |
| Muscle | 0.2008 [0.1959, 0.2058] | 6.38% |
| Brain | 0.1762 [0.1723, 0.1800] | 5.63% |
| Liver | 0.1750 [0.1613, 0.1888] | 20.64% |
| Bone | 0.1712 [0.1639, 0.1785] | 9.84% |
| Skin | 0.1434 [0.1373, 0.1495] | 9.38% |

Note: Plasma alanine (plaALA) TTR is the mean of each mouse's average plaALA TTR. Alanine TTR reported for each tissue is the intracellular alanine TTR (icALA TTR). Data are expressed as mean [95% CI]. n = 30 for plasma and tissues.

**Table 3. Protein-bound alanine (pbALA) TTR and comparison of protein FSR calculated with plasma or intracellular precursor pools for all mice combined.**

| Tissue | pbALA TTR | FSR (plasma) | FSR (intracellular) | P-value |
|---|---|---|---|---|
| Kidney | 0.0408 [0.0396, 0.0420] | 0.1809 [0.1769, 0.1848] | 0.2515 [0.2226, 0.2804] | <**0.0001**\* |
|  | 7.70% | 5.67% | 29.03% |  |
| Heart | 0.0163 [0.0146, 0.0181] | 0.0679 [0.0606, 0.0753] | 0.0792 [0.0717, 0.0868] | **0.0116**\* |
|  | 24.6% | 25.03% | 20.84% |  |
| Fat | 0.0280 [0.0280, 0.0315] | 0.1274 [0.1211, 0.1337] | 0.1352 [0.1281, 0.1422] | 0.1372 |
|  | 14.25% | 11.96% | 12.39% |  |
| Lung | 0.0164 [0.0156, 0.0193] | 0.0675 [0.0647, 0.0704] | 0.0796 [0.0735, 0.0857] | **0.0125**\* |
|  | 12.94% | 11.18% | 19.79% |  |
| Jejunum | 0.0736 [0.0705, 0.0767] | 0.3644 [0.3482, 0.3805] | 0.4780 [0.4298, 0.5263] | <**0.0001**\* |
|  | 10.99% | 11.66% | 26.53% |  |
| Ileum | 0.0644 [0.0610, 0.0679] | 0.3070 [0.2904, 0.3236] | 0.3601 [0.3246, 0.3956] | **0.0053**\* |
|  | 14.15% | 14.23% | 25.91% |  |
| Muscle | 0.0074 [0.0069, 0.0079] | 0.0298 [0.0277, 0.0320] | 0.0299 [0.0277, 0.0320] | 0.9378 |
|  | 16.83% | 17.43% | 17.06% |  |
| Brain | 0.0179 [0.0174, 0.0183] | 0.0741 [0.0726, 0.0756] | 0.0858 [0.0828, 0.0888] | <**0.0001**\* |
|  | 6.38% | 4.77% | 8.23% |  |
| Liver | 0.0579 [0.0558, 0.0600] | 0.2723 [0.2625, 0.2821] | 0.3269 [0.2935, 0.3603] | **0.0038**\* |
|  | 8.93% | 8.70% | 24.75% |  |
| Bone | 0.0076 [0.0065, 0.0087] | 0.0308 [0.0263, 0.0353] | 0.0365 [0.0310, 0.0419] | 0.1684 |
|  | 30.93% | 31.09% | 31.09% |  |
| Skin | 0.0050 [0.0041, 0.0059] | 0.0201 [0.0167, 0.0235] | 0.0277 [0.0231, 0.0322] | **0.0475**\* |
|  | 35.89% | 35.31% | 34.02% |  |

Note: FSR (plasma) indicates that plaALA TTR was used as the precursor pool during FSR calculation. FSR (intracellular) indicates that icALA TTR was used. p-value is reported for t-test between FSR (plasma) and FSR (intracellular). All FSR calculations used the same pbALA TTR. Data are expressed as mean [95% CI] and coefficient of variation(%). Protein FSR is fraction/day. n = 30 for all and a mix of high-active and low-active.

\*significance of unpaired t-test ($p < 0.05$)

Analysis of protein FSR per physical activity levels resulted in a significant protein FSR difference in the kidney tissue between the high-active and low-active mice (Fig 4). The kidney protein FSR in the low-active mice was 7.82% higher than the high-active mice. No significant differences were found between protein FSR of high-active and low-active mice in any of the other studied organs (heart, fat, lung, jejunum, ileum, liver, muscle, brain, bone, and skin).

## Discussion

The purpose of the present study was to determine the differences in protein fractional synthesis rates (FSR) of various organ tissues between high- and low-active mice, using $D_2O$. Only in kidney tissue, we identified a significant difference between protein FSR of high- and low-active mice suggesting a possible connection between renal protein synthesis and physical activity level.

### Fractional synthesis rate of high- and low-active mice

Protein FSR is an important measure to gain an understanding of metabolic regulation, as the faster the organ can synthesize protein, the more control the tissue has through upregulation and downregulation of metabolic pathways. It, therefore, represents the organ's ability to adapt to a change of environment. We calculated protein FSR of multiple organs in high-active

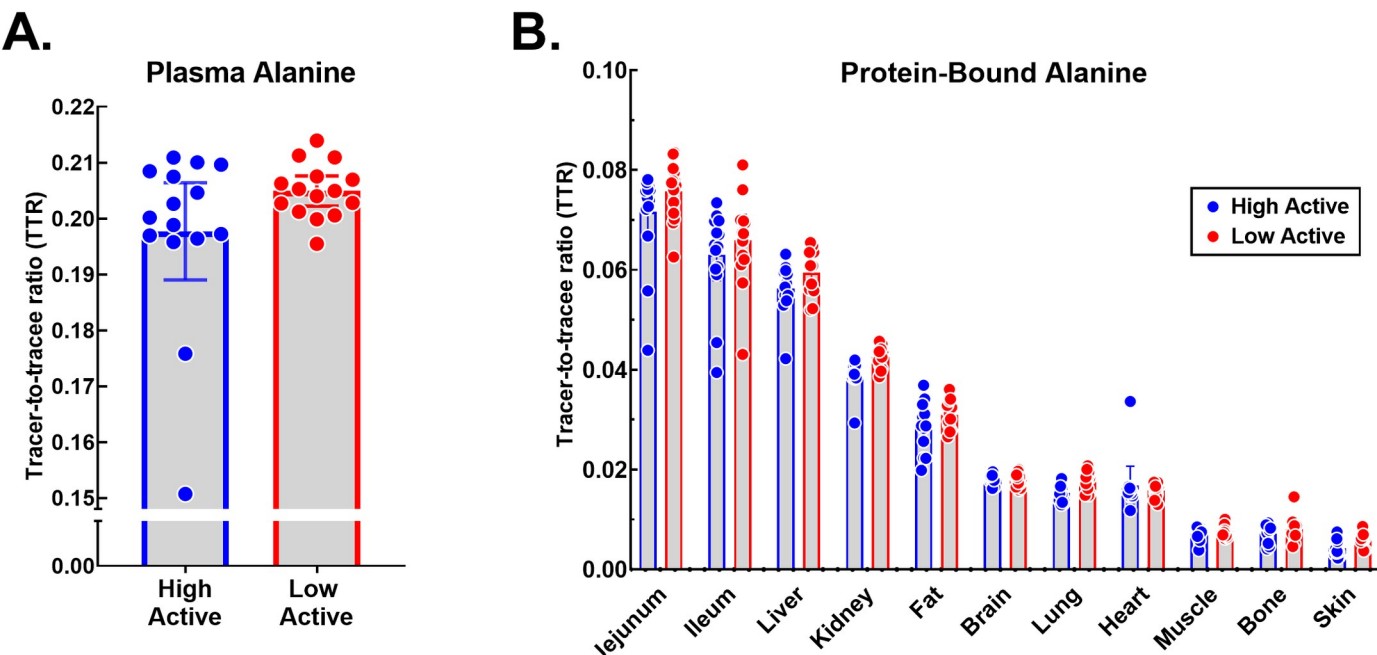

**Fig 2. Alanine TTR of high and low active mice.** (A) Plasma alanine TTR of high and low active mice. (B) Protein-bound alanine TTR of high and low active mice per organ tissue. All data are mean [95% CI], n = 15.

and low-active mice to analyze the relationship between tissue protein synthesis and physical activity level. Our previous studies have identified a link between metabolism and physical activity level [2, 3]. In contradiction to our hypothesis, we found that the kidney of the low-active mice had a 7.8% higher protein FSR than the high active mice, but no significant differences were observed in any of the other studied tissues. Since we used inbred strains with inherently differing physical activity levels, the results indicate that the higher FSR in the kidney of low active mice may be contributing to a change in physical activity level rather than a consequence of low activity level. Although an argument can be made that a 7.8% difference in protein synthesis may not be physiologically relevant in a tissue with a high protein FSR like the kidney, many other studies have observed associations between physical activity level and the progression of chronic kidney disease [27, 28]. Evidence also exists that the benefit of physical activity for the kidney lies in the correction of blood pressure, reduction of inflammation, decrease in triglycerides, and reduction of co-morbid diseases [28], but is not related to any tissue-specific metabolic changes of the kidney instead of systemic adaptations. No studies, to our knowledge, have examined the physical activity-induced physiological changes on kidney tissue metabolism. Without a solid understanding of the role that physical activity plays on kidney tissue metabolism, we can not definitively explain a connection of a higher kidney protein FSR and a lower physical activity level.

In addition, a statistical trend was observed in the lung tissue with higher protein FSR in the low-active mice when compared to high-active mice. It is known that the lung tissue protein synthesis decreases from gestation to adult life then remains constant throughout adult life [29]. As a sedentary lifestyle is mostly accompanied by a reduced ventilatory load, one might expect protein FSR of the lungs to be suppressed in low-active mice. On the other hand, lung tissue protein synthesis has been demonstrated to increase in damaged states [30]. Moreover, in physically inactive chronic bronchitis patients with enhanced pulmonary and systemic low grade inflammation, higher resting whole-body protein turnover was observed [31].

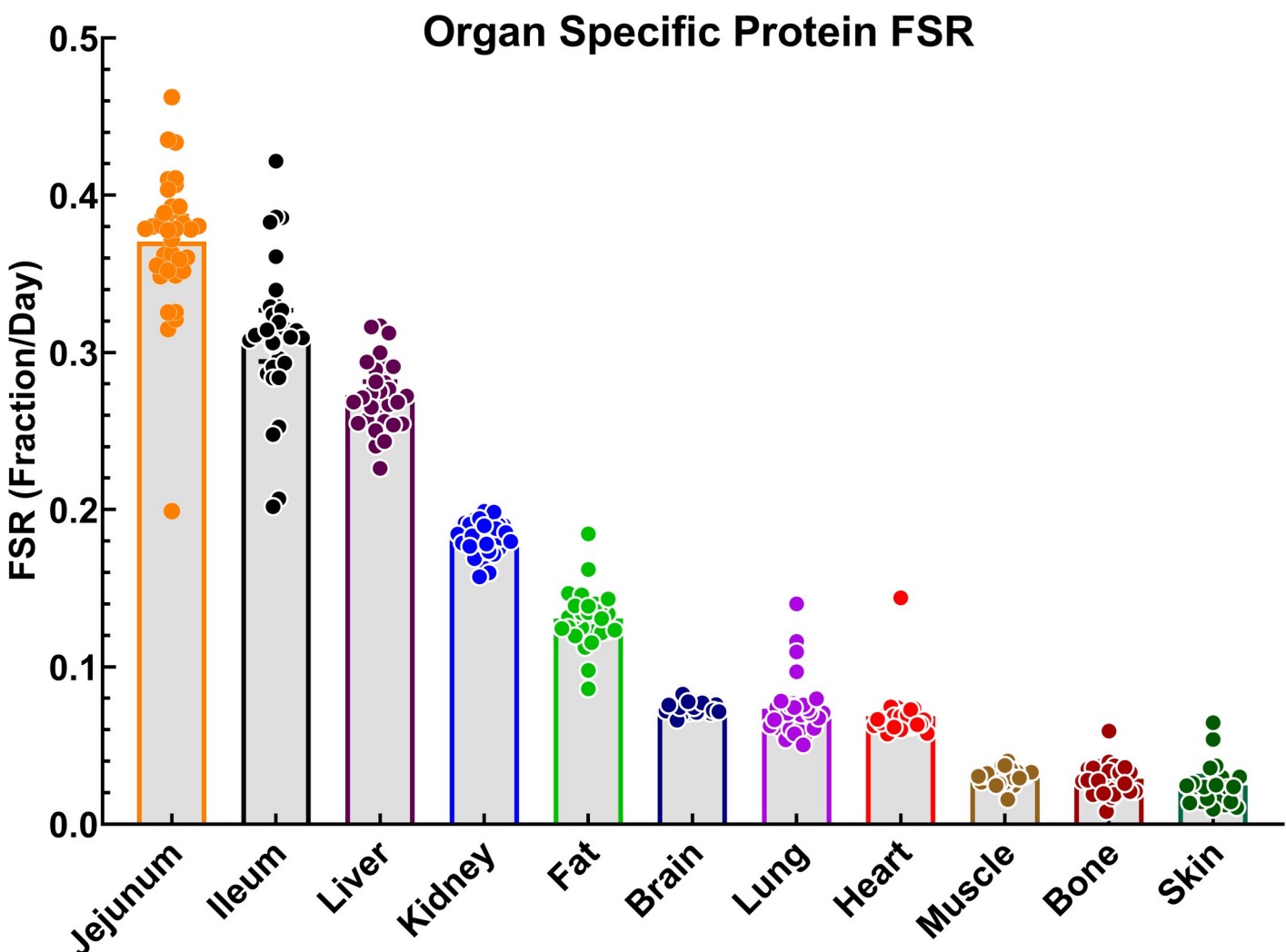

**Fig 3. Order of tissue protein FSR in various tissues from highest to lowest of combined high- and low-active mice.** All data are mean [95% CI], n = 30.

Whether a chronic low grade inflammatory state is also present in the low-active mice in these mice needs further study but could possibly contribute to the inability to achieve a higher physical activity level. More research is needed to examine if protein synthesis of the lung can be modified by a change in physical activity and whether the upregulated protein FSR of the lung is responsible for the inability to achieve a higher physical activity level.

Overall, the present study shows that most of the measured high-active mice organs are not differently equipped for adaptation through protein FSR rates than the low-active mice. These results suggest that the source of physical activity regulation is not an increase in protein turnover but that the low active mice lack the ability to increase protein FSR to achieve a higher physical activity level.

### Comparison of obtained data to literature

We compared the protein FSR in the tissues with those described in the literature and observed that the jejunum had the highest protein FSR followed by the liver, ileum, heart, and skeletal muscle in accordance with the reported protein FSR in multi-organ analyses [4, 7]. Our

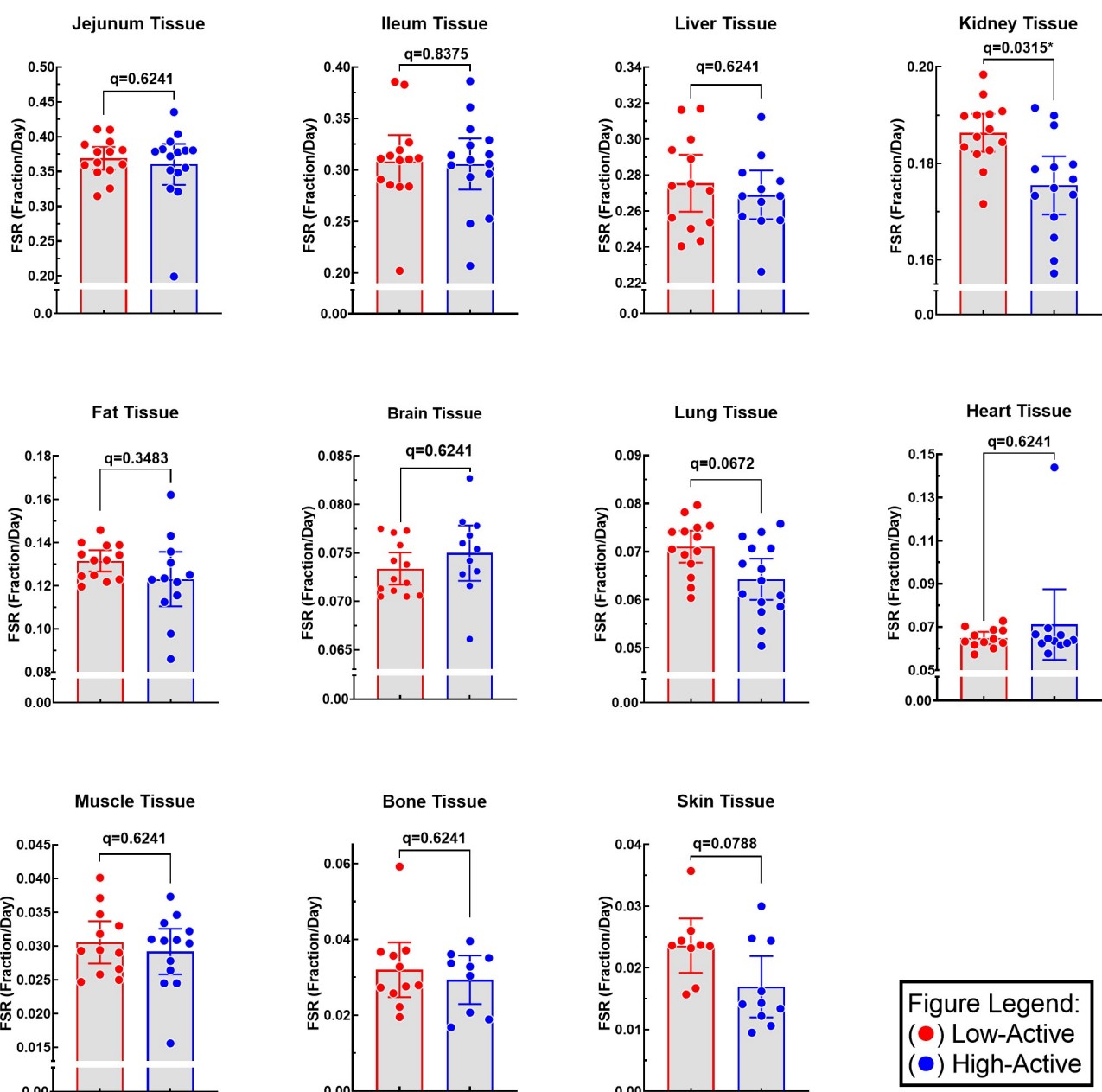

**Fig 4. Protein FSR of low-active and high-active mice in each tissue.** Statistics: t-test corrected with false discovery rate (significance: q<0.05). Data are expressed as mean [95% CI] in fraction/day.

observed tissue-specific protein FSR values were comparable to those reported using $D_2O$ [7, 20, 32] and [$^2H_5$]phenylalanine flood dose methods [33, 34] in mice [33], rat [7], and humans [20].

## Limitations

The mice used in this experiment were administered a pulse of stable isotope tracers an hour before tissue collection for analysis of amino acid whole body production and clearance as reported [17] and the mice underwent surgery with anesthesia. The anesthesia and surgical

procedure would not make a large impact on the protein FSR as the incorporation of tracer from $D_2O$ has taken place over 24 hours. Therefore, an increase or decrease in protein FSR due to the surgery would have only affected the last hour of a 24 hour measurement. A 20% reduction in the FSR during the last hour would only have about 1% effect on the measured FSR. In addition, all mice underwent the same type of surgery and we can anticipate that the effect on FSR would be roughly the same. There is a possibility that these stable isotope tracers increased the amount of isotopes ($^5N$, $^{13}C$ or $^2H$) available since the labeled isotopes of the administered amino acids can appear in alanine [35]. We used plasma to calculate protein FSR due to the icALA TTR variability that possibly could be caused by the stable isotope pulse. However, we acknowledge that the intracellular alanine probably is the most optimal choice of precursor pool, because this pool represents the labeled alanine that is directly used by the cell for protein synthesis. We could use plaALA TTR since the enrichment of alanine should be the same in all tissues of the mouse [25] and our calculations using plasma did not change our overall conclusions. In future studies, plasma should be taken before the administration of stable isotope tracer pulse to obtain the highest accuracy. We recommend if intracellular alanine is desired to be used as the precursor pool then the subject should not be administered other stable isotope tracers.

The observed significant role of the kidney tissue and lung tissue may be (in part) related to other potential strain differences. However, extensive research has been done on the genomic and proteomic profiles between these two inbred strains in order to understand the role of genetics on physical activity level difference [1, 12]. With genetic control, as these strains are inbred, and with environmental controls, such as diet, we are able to contribute findings to a difference in the metabolic pathway.

## Conclusion

In summary, the present study results show that physical activity level is not specifically regulated by protein FSR, although more in-depth analysis is required regarding the observed relationship between kidney tissue metabolism and physical activity. This work is important to determine which biological factors are regulating physical activity level, as physical inactivity is linked to several chronic diseases [36] and is a growing societal problem. More studies are needed to examine whether other metabolic parameters are able to differentiate the physical activity level. The methods used in the present study make it possible to explore the utilization of multiple energy substrates such as fatty acid oxidation and glycogen storage in the liver and muscle [6], whereas others also measured fatty acid and lipid synthesis using $D_2O$ [37, 38]. In addition, protein breakdown analysis of the different tissues would provide a more complete metabolic picture in regards to the physical activity level.

## Acknowledgments

The authors would like to acknowledge the students of Biology of Physical Activity Laboratory that helped with tissue collection and preparation: Ayland Letsinger, Chaz Nagel, Tatiana Podovani, Cristina Osorio, Jeremiah Velasco, and Victor Garcia. The study was designed by G.A.M.T.H., M.P.K.J.E., J.T.L., N.E.P.D.; data were collected and analyzed by K.M.C, J.Z.G, G. A.M.T.H, J.J.T; data interpretation and manuscript preparation were undertaken by K.M.C. Both J.T.L and N.E.P.D were joint senior authors. All authors approved the final version of the paper.

## Author Contributions

**Conceptualization:** Gabriella A. M. Ten Have, Marielle P. K. J. Engelen, J. Timothy Lightfoot, Nicolaas E. P. Deutz.

**Data curation:** Kristina M. Cross, Jorge Z. Granados, Gabriella A. M. Ten Have, John J. Thaden, Nicolaas E. P. Deutz.

**Formal analysis:** Kristina M. Cross, Jorge Z. Granados, Nicolaas E. P. Deutz.

**Investigation:** Jorge Z. Granados, Gabriella A. M. Ten Have, Marielle P. K. J. Engelen, J. Timothy Lightfoot, Nicolaas E. P. Deutz.

**Methodology:** Gabriella A. M. Ten Have, John J. Thaden, Marielle P. K. J. Engelen, J. Timothy Lightfoot, Nicolaas E. P. Deutz.

**Project administration:** Gabriella A. M. Ten Have, Marielle P. K. J. Engelen, J. Timothy Lightfoot, Nicolaas E. P. Deutz.

**Supervision:** Gabriella A. M. Ten Have, Marielle P. K. J. Engelen, J. Timothy Lightfoot, Nicolaas E. P. Deutz.

**Writing – original draft:** Kristina M. Cross.

**Writing – review & editing:** Jorge Z. Granados, Gabriella A. M. Ten Have, John J. Thaden, Marielle P. K. J. Engelen, J. Timothy Lightfoot, Nicolaas E. P. Deutz.

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
