## [Decision Letter · Decision Letter 0]

8 Sep 2020

PONE-D-20-23160

Protein fractional synthesis rates within tissues of high- and low-active mice

PLOS ONE

Dear Dr. Cross,

Thank you for submitting your manuscript to PLOS ONE. After careful consideration, we feel that it has merit but does not fully meet PLOS ONE’s publication criteria as it currently stands. Therefore, we invite you to submit a revised version of the manuscript that addresses the points raised during the review process.

In particular, both reviewers were concerned that important methods were excluded from this manuscript which, made the reported methods somewhat confusing and limited the interpretation of the results. The premise of the study is also confusing. Why do the authors feel that FSR drives physical activity and not the other way around and can this premise actually be tested with the current experimental design? In addition, many questions still remain on the determination of FSR in this study including the methods used, the natural differences between the two inbred strains, and why the plasma alanine was so high. Finally, the authors need to be sure that the format carefully follows the journals specific guidelines. Please address these and the other comments brought up by both reviewers. 

We look forward to receiving your revised manuscript.

Kind regards,

Randy Wayne Bryner, Ed.D.

Academic Editor

PLOS ONE

Journal Requirements:

Reviewers' comments:

Reviewer's Responses to Questions

**Comments to the Author**

1. Is the manuscript technically sound, and do the data support the conclusions?

Reviewer #1: Partly

Reviewer #2: Yes

2. Has the statistical analysis been performed appropriately and rigorously? 

Reviewer #1: I Don't Know

Reviewer #2: Yes

3. Have the authors made all data underlying the findings in their manuscript fully available?

Reviewer #1: Yes

Reviewer #2: Yes

4. Is the manuscript presented in an intelligible fashion and written in standard English?

Reviewer #1: No

Reviewer #2: Yes

5. Review Comments to the Author

Reviewer #1: In this manuscript Cross et al. report the fractional synthesis rate of different tissues in two inbred mouse strains with different levels of physical activity. The manuscript is poorly written and does not follow the journal’s formatting guidelines. I find the premise that FSR drives physical activity puzzling. It would be easier (and testable) to argue that physical activity impacts FSR. In addition to physical activity, the two inbred strains differ in other traits which may confound the interpretation of the results. It was unclear why FSR was determined after surgery until it was mentioned in limitations that these results were part of another study. The feed removal, the anesthesia and the surgical procedure may have affected FSR without offering any advantage to its determination. In addition the administration of fluids during surgery likely diluted the D2O enrichment. Furthermore, the D2O enrichment reported in table 2 (0.2416) is outside the 95% CI of the values reported in the text. In fact, this value seems rather high if the authors successfully achieved a 5% D2O enrichment. Some tables and figures repeat the same data (e.g., Table 4 and Fig. 2) and some other are redundant and unnecessary (e.g., Table 3 and Fig. 3).

Reviewer #2: Summary.

Experts in the application of tracer methods in studies of protein kinetics have examined whether physical activity is associated with fundamental differences in protein synthesis. Mice were studied using D2O and protein synthesis was measured in multiple tissues over a 24-hour period. Overall, there were no major differences except for kidney and a suggestion in lung. Although the data are largely negative (i.e. virtually no differences between groups), the study is generally well done but it seems like several choices in the protocol design should be better explained.

Major comments.

If words permit, you might add a sentence or 2 in the ABSTRACT regarding a high-level view of the data. For example, values of protein synthesis in different tissues follow an expected pattern.

The need for a surgery seems odd. It looks like you only collected blood? Is this necessary? Also, there’s a peculiar statement in the 1st sentence under LIMITATIONS – you note that “…a pulse of stable isotope tracers …” was given ~ 1 hour before samples were collected. It looks like you had more going on in this study than simply giving D2O. It would be helpful to clarify the true protocol. What pulse of tracers?

Table 1 shows values for “total-body water” – the values are < 2 ml. What does this represent – intake? Please clarify.

It’s unfortunate that you did not directly measure body water labeling. I can appreciate your attention towards the labeling of alanine in plasma vs respective tissues, indeed, alanine is the true precursor. That said, the plasma alanine labeling seems too high given the dose of tracer you’ve administered. For example, we might expect ~ 5% water enrichment which limits the alanine to ~ 18.5 or 20% (3.7 or 4 times), however, the plasma is ~ 24%. This seems too high, in fact, tissue enrichments are very much in line with the expectation of ~ 3.7-4x equilibration between water and alanine labeling. You might want to elaborate on this point. It’s also odd that the suppl data show plasma alanine ~ 20% in each group – how does each group reach ~ 20% in suppl data but the combined data in Table 2 seem to reach ~ 24%?

The legend for Table 2 and 3 should more clearly state that you combined data for all animals. It might be better to change the order of Figure 2 and 3. In Figure 3 you get at more general trends in data / differences across tissues (comparable to what you show in Table 2 and 3) and then break data into active vs inactive groups (e.g. Table 4 comes last, why not make the data in Figure 2 become Figure 3)?

Minor comments.

You might explain whether you think studying the animals during activity would have been better? For example, animals were effectively sedentary for ~ 2 weeks prior to the study. Why not study subgroups who are regularly running vs those who are not? Can you comment on food intake when animals are exposed to running? It looks like there would be more marked effects if one group normally runs ~ 0.6 km/day vs another runs ~ 9.5 km/day. Not asking to do this experiment, just bring readers along with the rationale. As well, you chose to study all animals in a fed state. Would it have been advantageous to randomize each group to overnight fed vs fasted protocols and look for phenotypes that way? Again, not asking for more studies, just outline your choices. You are addressing a very important problem (i.e. the interplay between lifestyle and metabolic regulation), the question/comment here is … would it be of value to probe this with a different experimental paradigm?

You might want to explain why different collision energies are used for M0 (-68 V) vs M1 (-29 V)alanine.

It looks like low resolution figures are used here, the graphics are “fuzzy”. Presumably you will need to modify in finalized versions.

It does not seem like you need suppl data, I would suggest you consider adding those data to the main paper.

6. PLOS authors have the option to publish the peer review history of their article (what does this mean?). If published, this will include your full peer review and any attached files.

Reviewer #1: No

Reviewer #2: No

---

## [Author Response · Author response to Decision Letter 0]

23 Oct 2020

Thank you for your time spent reviewing our manuscript and for your comments/suggestions. They were very helpful in making the manuscript stronger. Please see the uploaded file titled "response to reviewers" for responses to each comment. Thank you again.

---

## [Decision Letter · Decision Letter 1]

12 Nov 2020

Protein fractional synthesis rates within tissues of high- and low-active mice

PONE-D-20-23160R1

Dear Dr. Cross,

We’re pleased to inform you that your manuscript has been judged scientifically suitable for publication and will be formally accepted for publication once it meets all outstanding technical requirements.

Kind regards,

Randy Wayne Bryner, Ed.D.

Academic Editor

PLOS ONE

Additional Editor Comments (optional):

Reviewers' comments:

Reviewer's Responses to Questions

**Comments to the Author**

1. If the authors have adequately addressed your comments raised in a previous round of review and you feel that this manuscript is now acceptable for publication, you may indicate that here to bypass the “Comments to the Author” section, enter your conflict of interest statement in the “Confidential to Editor” section, and submit your "Accept" recommendation.

Reviewer #2: All comments have been addressed

2. Is the manuscript technically sound, and do the data support the conclusions?

Reviewer #2: Yes

3. Has the statistical analysis been performed appropriately and rigorously? 

Reviewer #2: Yes

4. Have the authors made all data underlying the findings in their manuscript fully available?

Reviewer #2: Yes

5. Is the manuscript presented in an intelligible fashion and written in standard English?

Reviewer #2: Yes

6. Review Comments to the Author

Reviewer #2: Thanks for taking the extra time to better explain the logic regarding the experimental design and for adding specific details regarding the protocol. The work should be easier for others to follow.

7. PLOS authors have the option to publish the peer review history of their article (what does this mean?). If published, this will include your full peer review and any attached files.

Reviewer #2: No

---

## [Editor Report · Acceptance letter]

17 Nov 2020

PONE-D-20-23160R1 

Protein fractional synthesis rates within tissues of high- and low-active mice 

Dear Dr. Cross:

I'm pleased to inform you that your manuscript has been deemed suitable for publication in PLOS ONE. Congratulations! Your manuscript is now with our production department. 

Kind regards, 

on behalf of

Dr. Randy Wayne Bryner 

Academic Editor

PLOS ONE